# The Dynamic Relationships between Educational Expectations and Science Learning Performance among Students in Secondary School and Their Later Completion of a STEM Degree

**DOI:** 10.3390/bs14060506

**Published:** 2024-06-18

**Authors:** Jerf W. K. Yeung

**Affiliations:** 1Department of Social and Behavioural Sciences, City University of Hong Kong, Hong Kong, China; ssjerf@gmail.com; 2Graduate School of Human Sciences, Osaka University, Osaka 565-0871, Japan

**Keywords:** educational expectations, science learning performance, STEM degree, progressive process

## Abstract

The educational expectations of students for themselves have been commonly corroborated to directly predict their higher academic performance. Nevertheless, some recent research has reported that the academic performance of students may also contribute to their better development of educational expectations. Moreover, more advanced but limited research has argued that both the educational expectations and academic performance of students are developmental and changeable over time rather than fixed and stable. Due to the importance of students’ science learning performance during the years of secondary school in relation to their later STEM development in adulthood, the current study is intended to investigate how the developmental and growth trajectories of students’ educational expectations and science learning performance reciprocally affect each other directly and inversely in secondary school and then later contribute to their successful completion of a STEM degree in adulthood. Based on the six-wave panel data containing a nationally representative sample of adolescent students from the Longitudinal Study of American Youth (LSAY), the current study was conducted by parallel-process latent growth curve modeling (PP-LGCM) and found that both the developmental and growth trajectories of students’ educational expectations and science learning performance were mutually predictive of each other across the years of secondary school, which then contributed to their later higher likelihood of successful completion of a STEM degree in adulthood. In addition, the conditional direct PP-LGCM model, which is to model the effects of students’ educational expectations on their science learning performance, and the conditional inverse PP-LGCM model, which is to model the effects of students’ science learning performance on their educational expectations, showed significant within- and cross-domain effects differently. The implications of the study findings related to educational reforms and policy designs are discussed.

## 1. Introduction

Educational researchers have long paid great attention to the importance of different structural and individual factors in contributing to the better development of students’ educational trace, among which educational expectations of students and their academic performance during the early years are commonly reported as important predictors of students’ later educational success [1,2,3]. It goes without saying that students who demonstrate better academic performance in secondary school may also perform better for their later enrollment in higher education and obtaining an undergraduate degree. This is important as the educational success of adolescent students in adulthood can be profoundly influential on their later health, income, and well-being across life trajectories [4,5,6,7]. Indubitably, acquiring an undergraduate degree at university can secure students a better career prospect, tangible economic gains, and more life chances in adulthood [8,9,10]. This is especially true for students who can successfully complete a major in science, technology, engineering, and mathematics (STEM), which is due to STEM graduates being recognized as the human power for the modernization and advancement of economic development, cultural transformation, societal innovation, and technological progression [11,12,13]. Nevertheless, although we know there is a positive relationship between educational expectations and academic performance among students [14,15], little is known about how the educational expectations of students for themselves may affect their science learning performance across secondary school years, which are both pivotal for them to enroll and complete a STEM degree in adulthood. This is meaningful as the declining rates of secondary school students selecting STEM majors and/or later successfully graduating with a STEM degree in higher education have been observed [16,17,18]. Apparently, the lowering numbers of university graduates who majored in the STEM field may impede the development of human modernization, eco-friendly technological advancement, and efficient economic growth [19,20], in which the latter may directly sway the human future. On the other hand, some limited short-term longitudinal research has reported that the relationship between educational motivation, e.g., academic engagement and self-efficacy, and students’ academic performance is reciprocally reinforced and influenced rather than unidirectionally from the former to the latter [21,22,23]. This denotes, in the current study, that the science learning performance of students may possibly shape students’ development and changes in educational expectations in secondary school, which then jointly contribute to students’ future educational achievement, such as successful graduation with a STEM degree in adulthood.

Although limited existing short-term longitudinal research of cross-lagged effect designs supported the bidirectional relationship between academic engagement or the self-efficacy and educational performance of students over time [21,23,24,25,26], these investigations can only reveal the temporal changes in correlations of the study relationships concerned, which are unable to tell how their developmental and growth trajectories may affect each other over time together and how these developmental and growth trajectories later contribute to students’ educational achievement in adulthood. Due to the paucity of research investigating how the dynamic and changeable processes of students’ educational motivation and academic performance may shape each other over time and then later contribute to the educational achievement of students, the current study is intended to examine the reciprocal relationship between the developmental and growth trajectories of students’ educational expectations and science learning performance in secondary school and analyze how they may jointly predict students’ later successful completion of a STEM degree in adulthood. Findings of this study can help unfold the long-term and complex reciprocity of students’ early educational motivation and academic performance in the transitional period of secondary school years in connection with their later educational success in adulthood. This is conducive to educational reforms and policy designs, especially for science education and students’ STEM development.

## 2. Theoretical Framework of the Study

### 2.1. The Direct Model from Educational Expectations of Students to Their Science Learning Performance and STEM Achievement

Cultivating STEM graduates and professionals is a progressive and cumulative process that is critically reliant on the formative years of secondary school for students to establish their educational motivation, e.g., educational expectations, and cultivate their essential science knowledge and skills adequately [11,27,28]. Consonantly, Mau, Domnick [29] once stated that “(e)ighth-grade students typically are in the crucial stage of exploring self and the world of work. Unlike some occupational fields, preparation for nontraditional occupations, especially in the areas of science and engineering, must begin early. …training in math and science needs to be sequential and uninterrupted from elementary school, and fundamentals must be mastered before high school” (p. 324). This is aligned with the expectancy value theory, which presumes that if students consider educational performance meaningful and valuable, they would have a greater expectancy and tendency to invest and input more efforts, time, and resources in their academic work, including learning science, to seek better grades and scores in school [30,31]. This is pivotally relevant to students’ later educational achievement in adulthood, e.g., graduation with a STEM degree. In fact, learning science knowledge and skills and performing better in science education are an incremental and stepwise progression that requires students to have strong educational motivation and commitment through propelling their efforts, determination, and intrapersonal and interpersonal resources for reaching higher academic outcomes [32,33,34,35]. This helps explain the importance of educational expectations in relation to students’ science learning performance and their later successful completion of a STEM degree in adulthood. Manifestly, students with higher educational expectations tend to have a stronger educational drive and academic engagement to pursue academic excellence. In this study, the educational expectations of students are defined by the Wisconsin model to represent students’ general academic aspirations and expectancy for working towards better educational performance and achievement [1,36]. Therefore, students’ educational expectations in this study refer to the expected educational level that a student believes and perceives to be attained and completed [2,36,37,38].

Empirically, Khattab and colleagues [39] found in their recent cross-sectional study that both educational expectations and aspirations of middle school students were significantly predictive of their higher school GPA, even adjusting for the students’ school attitude, parental expectations, and demographic covariates of gender, ethnicity, school type, and socioeconomic status. In another cross-sectional study, Ali et al. [40] used representative data from the 2018 Program for International Student Assessment (PISA) and found that the educational expectations of students were significantly predictive of their higher reading achievement in a representative sample of secondary school students and its subsamples of local Qatari, first-immigrant, and second-immigrant students. Furthermore, Park [41] conducted a longitudinal study and found that the economic deprivation of adolescent students measured in middle school significantly and negatively predicted the students’ completed years of education in adulthood, with the educational expectations of these students measured in high school significantly mediating the study relationship. Similarly, Yeung [42] recently corroborated that the individual- and school-level academic aspirations of immigrant students in middle school significantly predicted their individual academic aspirations in high school, which then together contributed to the immigrant students’ successful graduation with a four-year undergraduate degree in university.

Nevertheless, existing pertinent studies examining the relationships between students’ educational expectations and science learning performance or their later completion of a STEM degree are either cross-sectional or short-term longitudinal, focusing only on the time periods of primary school, middle school, or university education separately [43,44,45,46]. These investigations tell little about how the progressive and cumulative process of students’ educational expectations shapes their development and changes in science learning performance across the years of secondary school and then affects the students’ later educational achievement in adulthood, e.g., the successful completion of a STEM degree in adulthood. For example, in their cross-sectional study, Liu et al. [45] reported that self-educational aspiration, a single-item measure similar to educational expectations for achieving the highest level of education that the student expected to attain, was significantly and directly related to the better mathematics achievement of primary school students in China. Moreover, Larson et al. [46] found that the science interests and educational aspirations of first-year university students significantly predicted their later graduation with a degree majoring in science. This study reveals that students who graduated with a science major had significantly higher science interests and educational aspirations compared to their counterparts who graduated with nonscience majors. Apparently, the above-mentioned cross-sectional and short-term longitudinal studies did not consider that educational expectations of students are a developmental and changing process that can impact the development and changes of students’ science learning performance continuously and progressively over time, which may later jointly affect students’ successful graduation with a STEM degree in adulthood. Thereby, the current study attempted to longitudinally investigate how the developmental and growth trajectories of students’ educational expectations may positively lead to their developmental and growth trajectories of science learning performance in the years of secondary school, which then together contribute to students’ successful completion of a STEM degree in adulthood, a direct model named here for this proposed longitudinal causal relationship.

### 2.2. The Inverse Model from Science Learning Performance of Students to Their Educational Expectations and STEM Achievement

On the other hand, the relationship between educational expectations and the academic performance of students is not only unidirectional from the former to the latter but also reciprocal and inverse, which means that students of higher academic performance would propel them to keep a higher level of educational expectations for better academic outcomes continuously, a process that is called the ‘gain spiral’ [47]. Although existing research has reported the inverse relationship between certain concepts of students’ educational motivation and students’ academic performance [21,24,48,49], e.g., academic engagement and self-efficacy, only one pertinent longitudinal study has been found to investigate the reciprocal and inverse relationship between students’ educational expectations and educational performance. In his short-term longitudinal study, Zhang [2] employed a cross-lagged fixed-effect design to examine the bidirectional association between educational expectations and math scores in a sample of senior primary school students in China and found that the math achievement of students did significantly predict their later higher educational expectations, although the direct relationship from students’ educational expectations to their math achievement was insignificant. In fact, other longitudinal studies investigating the bidirectional association between students’ academic motivation and educational performance can provide relevant evidence for the possible inverse connection between educational expectations and the science learning performance of secondary school students proposed in the current study. In a three-year longitudinal study by Hughes et al. [24], they found that not only the educational engagement of primary school students significantly contributed to their subsequent reading and math scores from grade 1 to grade 3, but also that the reading and math scores of these students significantly predicted their higher educational engagement subsequently. In addition, Guo et al. [21] examined the longitudinal relationship between educational engagement and the reading achievement of students from preschool to grade 5 and found that students’ preschool and grade 3 reading achievement significantly and positively contributed to their educational engagement in grade 1 and grade 5, respectively. More recently, Hwang et al. [49] found that the academic performance of students in the first semester of grade 8 significantly and positively predicted their self-efficacy beliefs in the second semester of grade 8, and that the self-efficacy beliefs of students in the second semester of grade 8 also significantly and positively predicted their academic achievement in the first semester of grade 9.

The above-cited study findings are accordant with what Pekrun et al. [26] stated: “(p)erceived (academic) competence and control depend on students’ individual achievement history, with success strengthening control and failure undermining it (p. 1655).” Consonantly, the cognitive consistency perspective posits that individuals would strive to maintain consistency with the social identity and image established in their track records and past history, which indicates that people tend to “maintain a desired self-view or to communicate traits to others” through executing “context-appropriate action and the acquisition and validation of knowledge” to fit the role(s) for sustaining or boosting their positive self-view and traits [50]. Accordingly, the inverse relationship between educational expectations and the science learning performance of students is expected in the current study, meaning that the developmental and growth trajectories of students’ science learning performance would also positively shape the developmental and growth trajectories of students’ educational expectations in the years of secondary school, which then together predict their successful completion of a STEM degree in adulthood, an inverse model called here for this proposed longitudinal causal relationship.

### 2.3. The Present Study

More important and pertinently, some scholars have recently proposed to take a dynamic approach to studying the educational motivation, including educational expectations, and academic performance of students over time, to see how their developmental and changing processes affect students’ later educational achievement [4,22,51]. In fact, more advanced but limited longitudinal studies have recently reported that the educational motivation and academic performance of students are an adaptation process that is developmental and changeable over time rather than fixed and stable [4,52]. It is indubitable that secondary school is a critical, transformative period for students to cultivate and establish their academic interests and educational goals in preparation for higher education and career development in the future. For this reason, a dynamic approach should be adopted to study the relationship between the educational expectations and science learning performance of students in the years of secondary school and see how they may contribute to their later educational achievement in adulthood, e.g., the successful completion of a STEM degree [22,26]. For instance, O‘Donnell et al. [51] examined the intrapersonal changes in students’ educational expectations and school functioning in a random sample of Australian adolescent students across the four time points in the students’ ages of 12/13, 14/15, 16/17, and 18/19 years old and found that educational expectations and school functioning were reciprocally and significantly influential of each other, which then contributed to students’ later educational achievement measured by the Australian Tertiary Admission Rank (ATAR). As cited above, in their three-year longitudinal study of primary school students, Hughes et al. [24] reported that the subsequent development of educational engagement, a measure related to educational expectations, and the reading and math scores of these students were both affected by their prior educational engagement and reading and math scores from grade 1 to grade 3, respectively and mutually. This is consonant with the Bayesian learning theory, which states that students are not passive recipients of what they face but can act as proactive agents to cultivate and modify their education-related situations and behavioral presentations continuously and progressively in order to achieve what they have set as academically important and meaningful for their educational development [53,54]. Thereby, it is research-worthy in the current study to examine how the developmental and growth trajectories of students’ educational expectations and science learning performance in the years of secondary school mutually affect each other, which then contribute to their later successful graduation with a STEM degree in adulthood.

In testing the above-proposed longitudinal causal relationships dynamically, the developmental and growth trajectories of students’ educational expectations and science learning performance are considered as the developmental trend of a student relative to their classmates in a standing track in the years of secondary school (the development of educational expectations and science learning performance) and the intrapersonal changes of this student across the years of secondary school (the growth of educational expectations and science learning performance) captured latently by repeated measurements from grade 8 to grade 12. In addition, students’ successful completion of a STEM degree in adulthood indicates whether the student participant graduated with a four-year baccalaureate degree majoring in science, technology, engineering, mathematics, or medicine (STEMM) or not, which is a common definition adopted in STEM research [43,55]. The time frame of the current study stretches across the years of secondary school from grade 8 to grade 12 and then to the mid-thirties of the student participants to see if the development and changes in students’ educational expectations and science learning performance during secondary school may distally affect their later STEM achievement in mid-adulthood. This extensive and intensive investigation of the impacts of students’ educational expectations and science learning performance on students’ later STEM development outperforms existing relevant longitudinal research that mainly focused on a limited time period, e.g., primary school, middle and high school, or higher education [2,21,51,52]. This leaves the interconnected and extended relationship between students’ educational development in secondary school and their later STEM achievement in adulthood uncharted. Hence, the findings of the current study help answer a long-standing question regarding whether and how the processes of students’ educational expectations and science learning performance in secondary school may be distally related to the cultivation of future STEM talents. Taken together, the current study has the following hypotheses:

**H1:** 
*The developmental and growth trajectories of students’ educational expectations would positively affect the developmental and growth trajectories of students’ science learning performance during the years of secondary school (direct model).*


**H2:** 
*The developmental and growth trajectories of students’ science learning performance would positively affect the developmental and growth trajectories of students’ science learning performance during the years of secondary school (inverse model).*


**H3:** 
*The developmental and growth trajectories of students’ educational expectations and science learning performance during the years of secondary school would positively predict their successful completion of a STEM degree in adulthood (direct and inverse models).*


To test the above hypotheses, the sociodemographic variables of students’ gender, family composition, and ethnicity are controlled to preclude possible confounding effects in the modeling procedures. This is because female students are generally found to have higher educational motivation and academic performance than their male counterparts [18,35,56], although their different STEM intentions and enrolments in STEM majors reported by recent research indicate that female students are more fond of health-related and life science subjects while their male counterparts tend to be interested in physical and computer science [3,18,57]. In addition, students living with both their biological father and mother tend to have better educational expectations and academic achievement compared to their counterparts in other family structures [58,59]. In this study, student participants are classified into five ethnic groups that include Hispanics, African Americans, whites, Native Americans, and Asians, in which Asian students were set as the reference group due to their stronger educational drive and academic outperformance [60,61,62,63].

## 3. Methods

### 3.1. Sample and Data

The data for the current study are obtained from the Longitudinal Study of American Youth (LSAY), which is a nationwide and representative survey of middle and high school students studying in public schools in the United States [64]. LSAY employed a stratified sampling framework based on two sampling strata defined by the geographic regions and types of communities in the country, in which two cohorts of student participants were randomly drawn from 52 public secondary schools in the country. The first cohort sampled 2829 high school students in grade 10, and the second cohort recruited 3116 middle school students in grade 7, which were representative of the public middle- and high-school student populations in the United States. The longitudinal interviews of LSAY were conducted annually from 1987 to 1994 for 7 years: 3 years of high school and 4 years after high school for cohort-1 students; and 3 years of middle school, 3 years of high school, and 1 year after high school for cohort-2 students. In 2006, LSAY received additional funding from the National Science Foundation (NSF) to trace the educational and occupational development of cohort 1 and cohort 2 students in their adulthood, in which five additional longitudinal surveys were conducted in 2007, 2008, 2009, 2010, and 2011. In 2007, LSAY successfully relocated around 95% of the original sample of cohort 1 and 2 students (N = 5945), and these student participants were of average ages between 33 and 37 years old. The original sample consisted of approximately 48% female and 52% male students, 70% whites, 17% African Americans, 9% Hispanics, 3% Asian Americans, and 1% Native Americans. The current study only used data from cohort-2 students due to their containing the study variables across secondary school from grade 8 to grade 12 and adulthood.

### 3.2. Measures

Educational expectations of students were measured annually from grade 8 to grade 12 by an item provided by the personnel of LSAY: “As things stand now, how far in school do you think you will get?”. The item was measured on a 6-point scale: 1 = high school only, 2 = vocational/trade school, 3 = some college, 4 = bachelor’s degree, 5 = master’s degree, and 6 = doctorate/professional degree, in which higher scores represent higher educational expectations. This single-item measure aligns with the Wisconsin model for measuring educational expectations [15,36], which has been commonly used in empirical research to assess students’ educational expectations [2,36,38].

Science learning performance of students was measured by the averaged grade scores of the student’s science subjects of biology, chemistry, and physics from grade 8 to grade 12 annually, which were provided by the school personnel of respective participating schools in LSAY and assessed by an 8-point scale ranging from 1 = mostly A, 2 = half A and half B, 3 = mostly B, 4 = half B and half C, 5 = mostly C, 6 = half C and half D, 7 = mostly D, and 8 = mostly below D. For easy interpretation, the scale ratings were reversely coded, in which higher scores represent better science learning performance. Existing research has adopted this measurement approach to rate the science achievement of students [65,66].

The completion of a STEM degree in adulthood was measured in 2009 by an item provided by the personnel of LSAY, which was to indicate whether the student participant had successfully graduated with a four-year baccalaureate degree majoring in science, technology, engineering, mathematics, or medicine (STEMM) or otherwise. This dichotomous classification of graduates with a STEM degree compared to their counterparts who completed a non-STEM degree or had no baccalaureate degree has been used in existing empirical research [55,67], which is coded as follows: 0 = non-STEMM major or no baccalaureate, and 1 = STEMM major.

The sociodemographic variables to be controlled in the modeling procedures to test the hypotheses of the current study include gender, family composition, and ethnic origin, which were collected in grade 7. In this study, gender (0 = female, 1 = male) and family structure (0 = otherwise, 1 = two-parent family) are dichotomous variables. The ethnic origin of the students was classified into four dummy variables, with Asian students as the reference group (0) and white, African American, Hispanic, and Native American students as the comparison (1). The age of students was not included in the modeling procedures due to its alignment with the years of secondary school to avoid multicollinearity [68].

### 3.3. Modeling Techniques

To predict the outcome of students’ successful completion of a STEM degree in adulthood as a result of their developmental and growth trajectories of educational expectations and science learning performance in secondary school, parallel-process latent growth curve modeling (PP-LGCM) was conducted [69]. PP-LGCM is an extended type of growth modeling that is reliable and flexible for analyzing the longitudinal casual relationships of their latent trajectories in connection to a distal outcome. In the growth modeling framework, latent variables are estimated for the initial developmental levels (intercepts) and the changes (slopes) occurring over time that are used to represent the developmental and growth trajectories of students’ educational expectations and science learning performance during the years of secondary school from grade 8 to grade 12 in the current study. The conditional form of a PP-LGCM model can be written as follows:$$y_{ti}^{m}=\eta_{0_{i}}^{y_{m}}+\lambda_{t}^{y_{m}}\eta_{1_{i}}^{y_{m}}+\varepsilon_{t_{i}}^{y_{m}}\;{\;\eta }_{0_{i}}^{y_{m}}=\eta_{0}^{y_{m}}+\sum_{k}\gamma_{0_{k}}^{y_{m}}X_{k}+\varsigma_{0_{i}}^{y_{m}}\;\eta_{1_{i}}^{y_{m}}=\eta_{1}^{y_{m}}+\sum_{k}\gamma_{1_{k}}^{y_{m}}X_{k}+\varsigma_{1_{i,}}^{y_{m}}$$
where the first equation is the within-subject model with $y_{ti}^{m}$ representing the observed outcome of students’ successful completion of a STEM degree in adulthood, and the model parameters $\eta_{0_{i}}^{y_{m}}$ and $\eta_{1_{i}}^{y_{m}}$ indicating the intercept and slope growth factors for the developmental and growth trajectories of students’ educational expectations and science learning performance in the years of secondary school, and $\varepsilon_{t_{i}}^{y_{m}}$ known as the residual term. The second and third equations are the between-subject models, in which $\eta_{0}^{y_{m}}$ and $\eta_{1}^{y_{m}}$ denote the estimated overall initial level of the developmental trajectories and the average rate of changes of the growth trajectories for the educational expectations and science learning performance of students during the years of secondary school, respectively. In addition, $\gamma_{0_{k}}^{y_{m}}$ and $\gamma_{1_{k}}^{y_{m}}$ represent the parameters of a set of the students’ time-invariant sociodemographic covariates in terms of their gender, family composition, and ethnicity presented as $\sum X_{k}$, and $\varsigma_{0_{i}}^{y_{m}}$ and $\varsigma_{1_{i}}^{y_{m}}$ are the error terms.

In the modeling procedures, the univariate direct and inverse PP-LGCM models were first fitted to confirm the shape and variance of the developmental and growth trajectories of students’ educational expectations and science learning performance across the years of secondary school [70]. The univariate direct PP-LGCM model was used to set the developmental and growth trajectories of students’ educational expectations as the predictors of students’ developmental and growth trajectories of science learning performance, and the univariate inverse PP-LGCM model was used to predict students’ developmental and growth trajectories of educational expectations by their developmental and growth trajectories of science learning performance. After the confirmation of the univariate direct and inverse PP-LGCM models, the conditional direct and inverse PP-LGCM models with the distal outcome of students’ successful completion of a STEM degree in adulthood were tested [68,69], in which the longitudinal causal relationships specified by the above-mentioned three equations can be integrated in a unified equation formula:$$\eta_{Di}=a_{D0}+\beta_{1}\eta_{Ii}+\beta_{2}\eta_{Si}+\beta_{3}z_{i}+\xi_{Di},$$
in which $\eta_{Di}$ is the distal outcome of students’ successful completion of a STEM degree in adulthood, $\beta_{1}\eta_{Ii}$ and $\beta_{2}\eta_{Si}$ are the regression parameters of the intercept and slope factors of students’ educational expectations and science learning performance to represent their developmental and growth trajectories in secondary school, and $\beta_{3}z_{i}$ are the regression parameters of the adjusted sociodemographic covariates of students’ gender, family composition, and ethnicity, with $a_{D0}$ and $\xi_{Di}$ indicating the intercept of the distal outcome of students’ successful completion of a STEM degree and $\xi_{Di}$ referring to the person-specific difference between $\eta_{Di}$ and $a_{D0}$, respectively. This type of modeling procedures for the study of longitudinal causal relationships has been recently used in educational, developmental, and behavioral science research [71,72].

The modeling procedures were fit in Mplus 8.7 [73]. In order to account for the interdependence of the data structure of LSAY nested at the school level to adjust for standard errors and chi-square tests, the COMPLEX function was conducted with the CLUSTER design by setting <type = complex> in the modeling procedures [70]. The full information maximum likelihood (FIML) method was used to handle missing values for achieving better generalization [70,74], which is deemed a special multiple imputation (MI) technique for the estimation of missing values in power, bias, and efficiency based on available data by the likelihood function [74]. Model fit was evaluated by comparative fit index (CFI), root mean-square error of approximation (RMSEA), and standardized root mean-square residual (SRMR). Acceptable model fit is: CFI > 0.90, RMSEA < 0.08, and SRMR < 0.1; and excellent model fit is: CFI > 0.95, RMSEA < 0.06, and SRMR < 0.08 [68,70].

## 4. Results

The descriptive statistics of the sociodemographic covariates of students’ gender, family composition, and ethnic origins, and the study variables of students’ educational expectations, science learning performance, and the successful completion of a STEM degree are shown in Table 1. Females and males shared 48% (*n* = 1495) and 52% (*n* = 1621), respectively, and 87.1% of the students came from two-parent families (*n* = 2715) compared to 12.9% of their peer classmates from other family structures (*n* = 401). For ethnic origins, whites were the majority (*n* = 2169, 69.6%), followed by African Americans (*n* = 504, 16.2%), Hispanics (*n* = 284, 9.1%), Asians (*n* = 112, 3.5%), and Native Americans (*n* = 47, 1.5%). The mean scores of students’ educational expectations present a slightly descending pattern with an up-turn in the least year of secondary school: $\bar{x}$ = 4.005 in grade 8, and $\bar{x}$ = 3.886 in grade 9, $\bar{x}$ = 3.717 in grade 10, $\bar{x}$ = 3.612 in grade 11, and $\bar{x}$ = 3.731 in grade 12. In addition, the average grade levels of students’ science learning performance across the years of secondary school also display a similar changing pattern to that of educational expectations with an up-turn in grade 11 and grade 12: $\bar{x}$ = 5.997 in grade 8, $\bar{x}$ = 5.843 in grade 9, and $\bar{x}$ = 5.629 in grade 10, 5.672 in grade 11, and 5.840 in grade 12. Moreover, only 7.7% of the students (*n* = 239) obtained a four-year STEM baccalaureate degree in 2009.

Before conducting PP-LGCM modeling, correlations of the study variables of students’ educational expectations and science learning performance in secondary school and their successful completion of a STEM degree in adulthood were run. Table 2 shows that the same- and cross-domain correlations of the study variables were all significant at *p* < 0.001. The same-domain correlations of students’ educational expectations from grade 8 to grade 12 ranged from r = 0.595 to 0.834 with a decreasing pattern of intrapersonal stability observed, in which the one-year lagged correlations of students’ educational expectations were from r = 0.731 to 0.824, and the two-year and three-year lagged correlations were from r = 0.689 to 0.747, and r = 0.640 to 0.680, respectively. In addition, the same-domain correlations of students’ science learning performance from grade 8 to grade 12 ranged from r = 0.311 to 0.592, which also show the same decreasing pattern of intrapersonal stability as to the same-domain correlations of students’ educational expectations, in which the one-year lagged correlations of students’ science learning performance were from r = 0.461 to 0.614, and the two-year and three-year lagged correlations were from r = 0.402 to 0.501, and r = 0.343 to 0.409, respectively. In fact, the decreasing levels of the lagged correlations among the same-domain measures across time verify the validity of their variability [75,76]. Besides, the cross-domain correlations between students’ educational expectations and science learning performance across the years of secondary school ranged from r = 0.323 to 0.394, which are smaller than those of the same-domain correlations but had a consistent decreasing pattern of correlations akin to the same-domain correlations. The concurrent cross-domain correlations of students’ educational expectations with science learning performance ranged from r = 0.291 to 0.351, and the one-year, two-year, three-year, and four-year lagged correlations ranged from r = 0.295 to 0.359, r = 0.242 to 0.310, r = 0.228 to 0.268, and r = 0.204 and 0.223, respectively. For the correlations between students’ successful completion of a STEM degree in adulthood and their educational expectations and science learning performance across the years of secondary school, the point-biserial correlation coefficients ranged from r = 0.184 to 0.243.

Table 3 shows the results of the univariate direct PP-LGCM model investigating the effects of students’ educational expectations on students’ science learning performance in secondary school (PP-LGCM Model 1A), for which a very good model fit was obtained: CFI = 0.972; RMSEA = 0.062; SRMR = 0.048. Nevertheless, the modification indices suggested setting the covariances between the residuals of students’ grade 9 and grade 10 educational expectations, grade 9 and grade 10 science learning performance, and grade 10 and grade 11 science learning performance, in which an excellent model fit appeared: CFI = 0.984; RMSEA = 0.048; SRMR = 0.039. The intercept factor loadings of students’ educational expectations across the years of secondary school from grade 8 to grade 12 were λ = 0.868, 0.866, 0.888, 0.901, and 0.862, *p* < 0.001; and the slope factor loadings from grade 9 to grade 12 were λ = 0.151, 0.310, 0.473, and 0.603, *p* < 0.001. The intercept factor loadings of students’ science learning performance across the years of secondary school from grade 8 to grade 12 were λ = 0.714, 0.728, 0.727, 0.761, and 0.824, *p* < 0.001; and its slope factor loadings from grade 9 to grade 12 were λ= 0.177, 0.354, 0.556, and 0.803, *p* < 0.001. For the within-domain regression effects, the slope factors of students’ educational expectations and science learning performance were significantly predicted by their intercept factors, β = −0.247 and −0.451, *p* < 0.001, suggesting that the growth trajectories of students’ educational expectations and science learning performance during the years of secondary school were susceptible to the impacts of students’ initial development in educational expectations and science learning performance in secondary school. This means that students with a higher initial development of educational expectations and science learning performance appeared to have a slower rate of growth in educational expectations and science learning performance across the years of secondary school later. For the cross-domain regression effects, the intercept factor of students’ educational expectations was significantly and substantially predictive of the intercept factor of students’ science learning performance in a positive way, β = 0.633, *p* < 0.001, indicating that students with a higher developmental trajectory of educational expectations contributed to their better developmental trajectory of science learning performance in secondary school. Furthermore, the slope factor of students’ educational expectations significantly and positively predicted the slope factor of students’ science learning performance, β = 0.216, *p* < 0.001, meaning that students with a better growth trajectory of educational expectations contributed to their better growth in science learning performance across the years of secondary school.

Next, the conditional direct PP-LGCM model was examined to predict the distal outcome of students’ successful completion of a STEM degree in adulthood by the developmental and growth trajectories of students’ educational expectations and science learning performance in secondary school while simultaneously adjusting for the sociodemographic covariates of students’ gender, family composition, and ethnic origins (PP-LGCM Model 1B). Figure 1 shows the results of PP-LGCM Model 1B, in which an excellent model fit was obtained: CFI = 0.988; RMSEA = 0.031; SRMR = 0.024. Specifically, the within- and cross-domain regression effects for the relationships between the developmental and growth trajectories of students’ educational expectations and science learning performance, plus their intercept and slope factor loadings, were similar to those of PP-LGCM Model 1A. Notably, the intercept factors of students’ educational expectations and science learning performance were both significantly and positively predictive of students’ successful completion of a STEM degree in adulthood, β = 0.145 and 0.244, *p* < 0.001, indicating that a unit increase in the developmental trajectories of students’ educational expectations and science learning performance resulted in the higher odds of students’ later graduation with a STEM degree by 15.6% and 27.6%, respectively. Furthermore, the slope factors of students’ educational expectations and science learning performance were also significantly and positively predictive of students’ successful completion of a STEM degree in adulthood, β = 0.083 and 0.152, *p* < 0.001, meaning that a unit increase in the growth trajectories of students’ educational expectations and science learning performance contributed to their higher odds of graduation with a STEM degree in adulthood by 8.6% and 16.4%, respectively. For the effects of students’ sociodemographic covariates, male students, compared to their female classmates, significantly had the higher developmental trajectory of science learning performance, β = 0.080, *p* < 0.01, and were also more likely to graduate with a STEM degree, β = 0.077, *p* < 0.01, OR = 1.080 (refer to Table A1 in the Appendix A). Moreover, students living in a two-parent family tended to have a higher growth trajectory of educational expectations and a higher developmental trajectory of science learning performance, β = 0.054 and 0.053, *p* < 0.05, and were also more likely to graduate with a STEM degree, β = 0.032, *p* < 0.05, OR = 1.032. In addition, compared to their Asian classmates, Hispanic, African American, and Native American students had a significantly lower developmental trajectory of educational expectations, β = −0.204, −0.104, −0.165, and −0.084, *p* < 0.001, 0.001, 0.01, and 0.01, and science learning performance in secondary school, β = −0.113, −0.161, −0.109, and −0.075, *p* < 0.001, 0.001, 0.01, and 0.01. Furthermore, African American and Native American students, in contrast to their Asian counterparts, had a significantly lower growth trajectory of science learning performance during the years of secondary school, β = −0.157 and −0.059, p < 0.05.

On the other hand, the univariate inverse PP-LGCM model was conducted to test the effects of students’ developmental and growth trajectories of science learning performance on their developmental and growth trajectories of educational expectations in secondary school (PP-LGCM Model 2A). Table 4 shows that an excellent model fit was obtained: CFI = 0.974; RMSEA = 0.059; and SRMR = 0.044. Nevertheless, the modification indices recommended setting covariances between the residuals of students’ grade 9 and grade 10, as well as grade 10 and grade 11 science learning performance, for which a better model fit was obtained: CFI = 0.987; RMSEA = 0.043; and SRMR = 0.033. The intercept factor loadings of students’ science learning performance in secondary school from grade 8 to grade 12 were λ = 0.715, 0.727, 0.726, 0.759, and 0.823, *p* < 0.001, and the intercept factor loadings of students’ science learning performance from grade 8 to grade 12 were λ = 0.871, 0.891, 0.913, 0.918, and 0.879, *p* < 0.001. In addition, the slope factor loadings of students’ science learning performance across the years of secondary school from grade 9 to grade 12 were λ = 0.175, 0.349, 0.547, and 0.790, *p* < 0.001, and the slope factor loadings of students’ educational expectations were λ = 0.159, 0.326, 0.491, and 0.627, *p* < 0.001. For the within-domain regression effects, the intercept factors of students’ science learning performance and educational expectations were significantly predictive of its slope factors, respectively, β = −0.445 and −0.463, *p* < 0.001, which means that the growth trajectories of students’ science learning performance and educational expectations were influenced by their developmental trajectories over the years of secondary school. Specifically, students with a higher initial development of science learning performance and educational expectations in secondary school presented a slower rate of growth in science learning performance and educational expectations afterward. For the cross-domain regression effects, the intercept factor of students’ science learning performance was significantly and positively predictive of both the students’ intercept and slope factors of educational expectations, β = 0.589 and 0.370, *p* < 0.001, which means that students of higher initial development of science learning performance contributed to their better developmental and growth trajectories of science learning performance in the years of secondary school. In addition, the slope factor of students’ science learning performance was significantly and positively predictive of the students’ slope factor of educational expectations, β = 0.370, *p* < 0.001, which indicates that students of progressive improvement in science learning performance over time also promoted their continuing enhancement of educational expectations across the years of secondary school.

Furthermore, the conditional inverse PP-LGCM model was examined to predict the distal outcome of students’ successful completion of a STEM degree in adulthood (PP-LGCM Model 2B). An excellent model fit was obtained: CFI = 0.988; RMSEA = 0.030; SRMR = 0.023 (Figure 2). The within-domain regression effects and intercept factor loadings, as well as the slope factor loadings of students’ science learning performance and educational expectations, were similar to those of PP-LGCM Model 2A. For the cross-domain regression effects, the intercept factor of students’ science learning performance significantly and positively predicted students’ intercept and slope factors of educational expectations, β = 0.610 and 0.367, *p* < 0.001. In addition, the intercept factor of students’ science learning performance and the intercept and slope factors of students’ educational expectations were jointly and significantly predictive of students’ successful completion of a STEM degree in adulthood, β = 0.267, 0.129, and 0.074, *p* < 0.001, 0.001, and 0.01. This means that a unit increase in the developmental trajectory of students’ science learning performance and the developmental and growth trajectories of students’ educational expectations contributed to the higher odds of students’ later graduation with a STEM degree in adulthood by 30.6%, 13.7%, and 7.6%, respectively. Furthermore, the slope factor of students’ science learning performance was also significantly and positively predictive of the slope factor of students’ educational expectations, β = 0.148, *p* < 0.01, and the successful completion of a STEM degree, β = 0.161, *p* < 0.001, in which the latter means that a unit increase in the growth trajectory of students’ science learning performance contributed to their higher odds of the acquisition of a STEM degree in adulthood by 17.4%. Furthermore, some of the students’ sociodemographic effects on their developmental and growth trajectories of science learning performance and educational expectations, as well as the successful completion of a STEM degree, remain similar to those of PP-LGCM Model 1B (refer to Table A2 in the Appendix A for the details). Specifically, male students, compared to their female classmates, had a significantly higher developmental trajectory of science learning performance, β = 0.017, *p* < 0.1, and were also more likely to graduate with a STEM degree, β = 0.080, *p* < 0.05, OR = 1.080. Furthermore, students in two-parent families had a higher growth trajectory in educational expectations, β = 0.095, *p* < 0.01. Moreover, compared to their Asian classmates, Hispanic, African American, and Native American students had a significantly lower developmental trajectory of science learning performance, β = −0.283, −0.332, −0.225, and −0.091, *p* < 0.001, 0.001, 0.001, and 0.05.

## 5. Discussion

The current study is the first of its kind to investigate the reciprocal influences of students’ developmental and growth trajectories of educational expectations and science learning performance directly and inversely on each other during the years of secondary school and how they contribute to the students’ distal outcome of successful completion of a STEM degree over time in mid-adulthood. Due to the malleability and cultivability of educational expectations and science learning performance among students in the course of secondary school, appropriate educational designs and policy interventions should be implemented and promoted to help secondary school students develop and establish better educational expectations and academic performance in different science subjects [14,43,44]. Generally, existing cross-sectional and short-term longitudinal research tends to investigate the relationships between educational expectations and the academic development of students by using correlational or cross-lagged effect designs to treat their effects as fixed and stable [2,14,39,77]. These studies manifestly overlooked the developing and changing nature of students’ educational expectations and academic development over time. In addition, little investigation has examined the relationships between students’ educational expectations and academic development, e.g., science learning performance, over the years of secondary school in relation to their later educational achievement in mid-adulthood, e.g., successful graduation with a STEM degree. Furthermore, as science education and the promotion of students’ STEM development are of interest to educators and policymakers because of their profound implications for the cultivation of STEM intellectuals to promote societal, economic, cultural, and technological advancements [12,28,55], more research is needed to scrutinize how the developmental and growth trajectories of students’ educational expectations and science learning performance during the years of secondary school may affect students’ later STEM success in adulthood. In the current study, both the conditional direct and inverse PP-LGCM models showed that the educational expectations and science learning performance of students impacted each other reciprocally in a direct and inverse way, which then jointly predicted students’ successful completion of a STEM degree 15 years later, in mid-adulthood. This reveals the importance of continuity and mutuality in the relationship between students’ academic motivation and educational development in the early years and their profound impacts carrying over on their later academic achievement extensively over time.

One of the main findings in the current study is that the developmental and growth trajectories of students’ educational expectations and science learning performance during the years of secondary school significantly and positively reinforce each other and then contribute to students’ later completion of a STEM degree in adulthood. For the developing and changing processes of students’ educational expectations and science learning performance in secondary school, the current study found that the growth trajectories of students’ educational expectations and science learning performance across the years of secondary school are susceptible to their own initial developmental trajectories in secondary school. This is congruent with a few more advanced longitudinal studies that point to the progressive nature of students’ academic motivation and educational development, formulated in the early years of secondary school [22,23,48]. As such, educators and adolescent practitioners should pay attention to the promotion strategies and pro-educational environments that can assist students to cultivate and enhance their educational expectations and science learning continuously and progressively in the course of secondary school, as these are profoundly influential on their later STEM achievement in adulthood [78,79]. The findings of the current study are consonant with existing longitudinal research with cross-lagged designs that the developmental and changing processes of students’ academic motivation and educational performance are mutually reinforced [26,48,80,81]. Nevertheless, the superiority of the current study compared to the existing longitudinal research studies of cross-lagged designs is that it takes a longer time span to examine the development and changes of students’ educational expectations and science learning performance across the years of secondary school and treats them as the latent developmental and growth trajectories to predict students’ later STEM achievement in adulthood. Hence, the findings of the current study give insights into the need for providing students with lower development of educational expectations and science learning performance, in the initial stage of secondary school, a more responsive and supportive educational initiative to avoid their subsequently accumulated academic lag due to their detrimental impacts on students’ STEM development in adulthood. In fact, the reciprocal relationships between the developmental and growth trajectories of students’ educational expectations and science learning performance supported in the current study reveal their cultivability and malleability, which should be targeted for educational innovations and policy reforms to help establish a stimulating and positive learning context to promote students’ educational expectations and science learning performance to better prepare them for later STEM achievement.

Moreover, the reciprocal relationship between the developmental and growth trajectories of students’ educational expectations and science learning performance is corroborated in the current study, in which students’ educational expectations and science learning performance shaped each other in a positive way over the course of secondary school. This is proved in the conditional direct and inverse PP-LGCM models (PP-LGCM Model 1B and PP-LGCM Model 2B), which reveal the importance of promoting students’ educational expectations and science learning performance simultaneously due to the bidirectionality between them. Nevertheless, stronger parallel cross-domain regression effects from students’ educational expectations to science learning performance were observed in both the univariate and conditional direct PP-LGCM Models (PP-LGCM Model 1A and 1B) in contrast to the inverse PP-LGCM Models (PP-LGCM Model 2A and 2B). This means that the effects of students’ developmental and growth trajectories of educational expectations on their developmental and growth trajectories of science learning performance compared to the inverse effects of students’ developmental and growth trajectories of science learning performance on their developmental and growth trajectories of educational expectations are manifestly different, such as β = 0.622 vs. 0.615 for the developmental trajectories and β = 0.215 vs. 0.148 for the growth trajectories. This explicates the robustness of the development and growth of students’ educational expectations in contribution to the development and growth of their science learning performance across the years of secondary school. Furthermore, only the transitioning cross-domain regression effect from students’ science learning performance to their educational expectations was significant in the conditional inverse PP-LGCM Model (PP-LGCM Model 2B), which sheds light on the importance of supporting students to perform better academically in secondary school as a learning drive to boost their growth of educational expectations that, in turn, contribute to their growth of science learning performance and successful graduation with a STEM degree in adulthood.

In addition, the current study found that the developmental and growth trajectories of students’ educational expectations and science learning performance were significantly predictive of the higher odds of students’ completion of a STEM degree in adulthood, which indicates that maintaining both high educational expectations and science learning performance and improving them progressively are pivotal for students to later obtain a STEM degree. Furthermore, the effects of students’ developmental and growth trajectories of science learning performance on their successful graduation with a STEM degree are explicitly stronger than those of the students’ developmental and growth trajectories of educational expectations in both the conditional direct and inverse PP-LGCM Models (PP-LGCM Model 1B and PP-LGCM Model 2B), which reveal that keeping up science learning performance at a higher level and enhancing it persistently over the years of secondary school are crucial for students to have better STEM achievement in adulthood.

On the other hand, the effects of the sociodemographic covariates of students’ gender, family composition, and ethnic origins on their developmental and growth trajectories of education expectations and science learning performance in secondary school and the completion of a STEM degree in adulthood are noteworthy. First, male students, compared to their female classmates, had a higher developmental trajectory of science learning performance and were more likely to graduate with a STEM degree, which induces a concern for how to promote science education and STEM development among the population of female students, although we know that female students are reported to have higher general educational performance [35,36]. In addition, the advantages of the better development of science learning performance and the growth of educational expectations are observed for students in two-parent families compared to their classmates in other family structures. It is important for educators and adolescent practitioners to create a pro-learning environment for students from disadvantaged family backgrounds to promote their academic motivation, science learning interests, and later STEM development [82,83]. Moreover, Hispanic, African American, white, and Native American students, compared to their Asian classmates, are found in the current study to generally have lower developmental trajectories of educational expectations and science learning performance in the years of secondary school, which may adversely impact their later STEM achievement in adulthood. This may be due to stronger family socialization for academic outperformance in Asian culture [84,85]. Nevertheless, more research is needed to scrutinize these ethnic differences related to educational motivation and STEM attainments.

## 6. Conclusions

The current study supported the argument that students’ educational expectations and their science learning performance in the course of secondary school education are developmental and changing in nature, in which the growth trajectories of students’ educational expectations and science learning performance are a function of their developmental trajectories in the same domain, and they are also mutually reinforced across time to contribute to students’ later successful completion of a STEM degree in adulthood. The above-mentioned longitudinal causal relationships between the developmental and growth trajectories of students’ educational expectations and science learning performance and students’ later STEM achievement are vindicated even after adjusting for the influential sociodemographic covariates of students’ gender, family composition, and ethnicity. In fact, the external validity of the findings in the current study is fortified by accounting for the nestedness of the clustered data structure of LSAY at the school level. Apparently, this study supported that both the educational expectations and science learning performance of students during the years of secondary school are developmental and changeable, which means that these important educational elements of students in the early years are cultivable and malleable. Educators and policymakers should hence support adolescent students to establish and maintain their higher educational expectations and better science learning performance in an ongoing process. This is important as they both have an effect on whether the students can succeed in STEM development in adulthood. In addition, the reciprocal and bidirectional relationship between students’ educational expectations and science learning performance was corroborated in the current study. Hence, the academic motivation of students and their educational performance during secondary school years should be monitored and supported by teachers, educators, and youth practitioners to help them have better development and sustenance in parallel because they are influential on students’ trajectories to become STEM professionals in the future. Furthermore, as the sociodemographic covariates of students, e.g., gender, family composition, and ethnicity, affect their developmental and growth trajectories of educational expectations and science learning performance, as well as STEM achievement in adulthood, responsive and effective educational and intervention strategies and initiatives should be introduced to support the STEM development of students with different sociodemographic characteristics.

Nevertheless, there are some limitations existing in the current study that need to be improved in future research. First, the data from LSAY were collected thirty years ago, starting in 1987 and ending in 2011, for which the learning attitudes and science education have apparently changed [12]. Hence, newer and updated data are needed to further confirm the study relationships found in the current study. Second, the sample of secondary school students from LSAY was recruited only from public schools, which excluded their counterparts studying in private schools. Therefore, the bidirectionality of students’ developmental and growth trajectories in educational expectations and science learning performance and their effects on students’ later completion of a STEM degree should be further verified in more representative data that include both students in public and private schools. Furthermore, future research should incorporate the multiple contextual systems of students’ family socialization related to education, peers’ learning behaviors, school climates and environments, and education-related supports at the neighborhood level as exogenous structural factors to see how they contextually shape the developmental and growth trajectories of students’ educational expectations and science learning performance during secondary school years in relation to their later graduation with a STEM degree in adulthood [11,12]. Lastly, although secondary school is a transitional stage for students to develop and establish their academic motivation and educational development, the learning attitudes, academic interests, and students’ foundational knowledge are formulated earlier in the period of primary school [54,86]. For this, future research should extend the time span to trace the development and changes of students’ educational expectations and their science learning performance across their whole primary and secondary school stages progressively and waveringly, only through which a clearer picture of students’ STEM development in adulthood can be reached.

## Figures and Tables

**Figure 1 behavsci-14-00506-f001:**
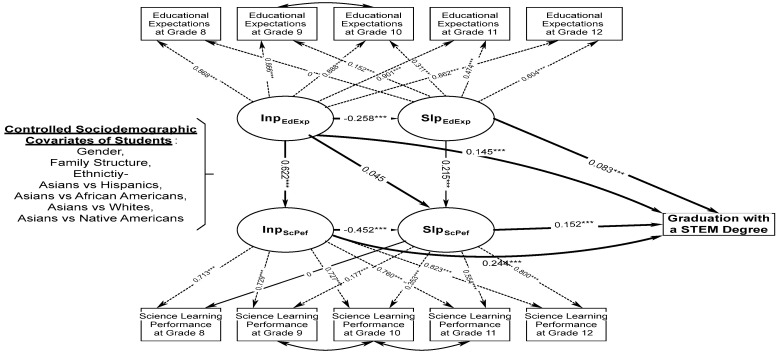
Parameter Estimates of the Conditional Direct Parallel Process Latent Growth Curve Model Predicting Students’ Developmental and Growth Trajectories of Science Learning Performance and Successful Completion of a STEM Degree by Their Developmental and Growth Trajectories of Educational Expectations Adjusting for the Sociodemographic Covariates of Students (PP-LGCM Model 1B). Note: For model fit, CFI = 0.988; RMSEA = 0.031; SRMR = 0.024. InpEdExp = Intercept Factor of Students’ Educational Expectations; SlpEdExp = Slope Factor of Students’ Educational Expectations; InpScPef = Intercept Factor of Students’ Science Learning Performance; SlpScPef = Slope Factor of Students’ Science Learning Performance. For simplicity, the regression effects of students’ sociodemographic covariates on students’ developmental and growth trajectories of educational expectations and science learning performance as well as successful graduation with a STEM degree are omitted; refer to Table A1 in the Appendix A for details. * *p* < 0.05, ** *p* < 0.01, *** *p* < 0.001.

**Figure 2 behavsci-14-00506-f002:**
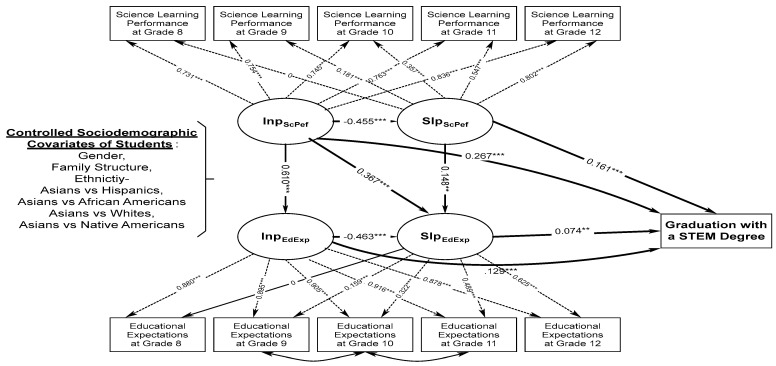
Parameter estimates the conditional inverse parallel process latent growth curve model predicting students’ developmental and growth trajectories of educational expectations and the successful completion of a STEM degree by their developmental and growth trajectories of science learning performance adjusting for the sociodemographic covariates of students (PP-LGCM Model 2B). Note: For model fit, CFI = 0.988; RMSEA = 0.030; SRMR = 0.023. InpEdExp = Intercept Factor of Students’ Educational Expectations; SlpEdExp = Slope Factor of Students’ Educational Expectations; InpScPef = Intercept Factor of Students’ Science Learning Performance; SlpScPef = Slope Factor of Students’ Science Learning Performance. For simplicity, the regression effects of students’ sociodemographic covariates on students’ developmental and growth trajectories of science learning performance and educational expectations, as well as successful graduation with a STEM degree are omitted; refer to Table A2 in the Appendix A for details. * *p* < 0.05, ** *p* < 0.01, *** *p* < 0.001.

**Table 1 behavsci-14-00506-t001:** Descriptive Statistics of the Sociodemographic and Study Variables of the Sample (*N* = 3116).

1.	Gender	Mean/Frequency	SD/%
	Female	1495	48%
	Male	1621	52%
2.	Family composition		
	Two-biological-parent	2715	87.1%
	Other	401	12.9%
3.	Ethnicity		
	White	2169	69.6%
	African American	504	16.2%
	Hispanic	284	9.1%
	Asian	112	3.6%
	Native American	47	1.5%
4.	Educational Expectations		
	Grade 8	4.005	1.846
	Grade 9	3.886	1.458
	Grade 10	3.717	1.419
	Grade 11	3.692	1.397
	Grade 12	3.731	1.380
5.	Science Learning Performance		
	Grade 8	5.997	1.734
	Grade 9	5.843	1.773
	Grade 10	5.629	1.801
	Grade 11	5.672	1.690
	Grade 12	5.840	1.515
6.	STEM Degree		
	Yes	239	7.7%
	No	2877	92.3%

Note: The sociodemographic variables of students’ gender, family composition, and ethnicity were collected in grade 7 in the year of 1987/1988; the educational expectations and science learning performance of the students were measured annually from grade 8 to grade 12 in the years of 1988/1989 to 1992/1993; and the outcome variable of students’ successful completion of a STEM degree was collected in the year of 2009.

**Table 2 behavsci-14-00506-t002:** Correlations of the Study Variables.

		1	2	3	4	5	6	7	8	9	10	11
1	Grade 8 EdExp	--										
2	Grade 9 EdExp	0.731	--									
3	Grade 10 EdExp	0.689	0.792	---								
4	Grade 11 EdExp	0.640	0.723	0.803	--							
5	Grade 12 EdExp	0.595	0.680	0.747	0.824	--						
6	Grade 8 ScPef	0.351	0.344	0.377	0.370	0.355	--					
7	Grade 9 ScPef	0.341	0.370	0.384	0.379	0.364	0.461	--				
8	Grade 10 ScPef	0.310	0.359	0.394	0.394	0.372	0.402	0.614	--			
9	Grade 11 ScPef	0.268	0.286	0.337	0.346	0.345	0.343	0.466	0.592	--		
10	Grade 12 ScPef	0.223	0.243	0.286	0.295	0.291	0.311	0.409	0.501	0.587	--	
11	STEM Degree	0.202	0.204	0.228	0.242	0.243	0.184	0.208	0.214	0.215	0.234	--

Note: EdExp = educational expectations; ScPef = science learning performance. STEM degree is dichotomously coded (0 = non-STEMM major or no baccalaureate, 1 = STEMM major), in which the correlations of STEM degree with other variables are point-biserial correlations. All correlational coefficients are significant at *p* < 0.001.

**Table 3 behavsci-14-00506-t003:** Parameter Estimates of the Univariate Direct Parallel Process Latent Growth Curve Model Predicting Students’ Developmental and Growth Trajectories of Science Learning Performance by Their Developmental and Growth Trajectories of Educational Expectations (PP-LGCM Model 1A).

**Intercept Factor Loadings**	**ϒ**	**SE**	**95% CI**
Inp_EdExp_			
Grade 8	0.868 ***	0.009	0.850 to 0.883
Grade 9	0.866 ***	0.011	0.864 to 0.906
Grade 10	0.888 ***	0.011	0.886 to 0.931
Grade 11	0.901 ***	0.015	0.884 to 0.942
Grade 12	0.862 ***	0.019	0.838 to 0.911
Inp_ScPef_			
Grade 8	0.714 ***	0.018	0.674 to 0.745
Grade 9	0.728 ***	0.020	0.687 to 0.765
Grade 10	0.727 ***	0.025	0.662 to 0.774
Grade 11	0.761 ***	0.026	0.705 to 0.809
Grade 12	0.824 ***	0.030	0.760 to 0.878
**Slope Factor Loadings**	**ϒ**	**SE**	**95% CI**
Slp_EdExp_			
Grade 8	--	--	--
Grade 9	0.151 ***	0.004	0.143 to 0.160
Grade 10	0.310 ***	0.009	0.293 to 0.329
Grade 11	0.473 ***	0.014	0.443 to 0.496
Grade 12	0.603 ***	0.018	0.565 to 0.634
Slp_ScPef_			
Grade 8	--	--	--
Grade 9	0.177 ***	0.008	0.162 to 0.192
Grade 10	0.354 ***	0.017	0.322 to 0.387
Grade 11	0.556 ***	0.024	0.508 to 0.602
Grade 12	0.803 ***	0.037	0.727 to 0.872
**Regression Effects**	**β**	**SE**	**95% CI**
Within-Domain Effects			
Inp_EdExp_ -> Slp_EdExp_	−0.247 ***	0.034	−0.335 to −0.200
Inp_ScPef_ -> Slp_ScPef_	−0.451 ***	0.058	−0.563 to −0.337
Cross-Domain Effects			
Inp_EdExp_ -> Inpt_ScPef_	0.633 ***	0.025	0.563 to 0.661
Inp_EdExp_ -> Slp_ScPef_	0.044	0.047	−0.046 to 0.137
Slp_EdExp_ -> Slp_ScPef_	0.216 ***	0.028	0.163 to 0.274

Note: For model fit, CFI = 0.984; RMSEA = 0.048; SRMR = 0.039. Inp_EdExp_ = Intercept Factor of Students’ Educational Expectations; Slp_EdExp_ = Slope Factor of Students’ Educational Expectations; Inp_ScPef_ = Intercept Factor of Students’ Science Learning Performance; Slp_ScPef_ = Slope Factor of Students’ Science Learning Performance. * *p* < 0.05, ** *p* < 0.01, *** *p* < 0.001.

**Table 4 behavsci-14-00506-t004:** Parameter Estimates of the Univariate Inverse Parallel Process Latent Growth Curve Model Predicting Students’ Developmental and Growth Trajectories of Educational Expectations by Their Developmental and Growth Trajectories of Science Learning Performance (PP-LGCM Model 2A).

**Intercept Factor Loadings**	**ϒ**	**SE**	**95% CI**
Inp_ScPef_			
Grade 8	0.715 ***	0.016	0.683 to 0.746
Grade 9	0.727 ***	0.019	0.690 to 0.764
Grade 10	0.726 ***	0.023	0.680 to 0.772
Grade 11	0.759 ***	0.026	0.708 to 0.809
Grade 12	0.823 ***	0.030	0.764 to 0.881
Inp_EdExp_	ϒ	SE	95% CI
Grade 8	0.871 ***	0.009	0.853 to 0.888
Grade 9	0.891 ***	0.011	0.870 to 0.911
Grade 10	0.913 ***	0.012	0.891 to 0.936
Grade 11	0.918 ***	0.015	0.889 to 0.947
Grade 12	0.879 ***	0.019	0.842 to 0.916
**Slope Factor Loadings**	**ϒ**	**SE**	**95% CI**
Slp_ScPef_			
Grade 8	--	--	--
Grade 9	0.175 ***	0.007	0.162 to 0.188
Grade 10	0.349 ***	0.014	0.320 to 0.377
Grade 11	0.547 ***	0.020	0.506 to 0.587
Grade 12	0.790 ***	0.034	0.724 to 0.857
Slp_EdExp_			
Grade 8	--	--	--
Grade 9	0.159 ***	0.004	0.150 to 0.167
Grade 10	0.326 ***	0.009	0.308 to 0.344
Grade 11	0.491 ***	0.013	0.466 to 517
Grade 12	0.627 ***	0.017	0.592 to 0.660
**Regression Effects**	**β**	**SE**	**95% CI**
Within-Domain Effects			
Inp_ScPef_ -> Slp_ScPef_	−0.445 ***	0.036	−0.515 to −0.375
Inp_EdExp_ -> Slp_EdExp_	−0.463 ***	0.041	−0.544 to −0.382
Cross-Domain Effects			
Inp_ScPef_ -> Inp_EdExp_	0.589 ***	0.021	0.549 to 0.629
Inp_ScPef_ -> Slp_EdExp_	0.370 ***	0.043	0.286 to 0.454
Slp_ScPef_ -> Slp_EdExp_	0.162 **	0.038	0.087 to 0.238

Note. For model fit, CFI = 987; RMSEA = 0.043, SRMR = 0.033. Inp_ScPef_ = Intercept Factor of Students’ Science Learning Performance; Slp_ScPef_ = Slope Factor of Students’ Science Learning Performance; Inp_EdExp_ = Intercept Factor of Students’ Educational Expectations; Slp_EdExp_ = Slope Factor of Students’ Educational Expectations. * *p* < 0.05, ** *p* < 0.01, *** *p* < 0.001.

## Data Availability

The data presented in this study are openly available in the Inter-university Consortium for Political and Social Research, which can be accessible at https://www.icpsr.umich.edu/web/ICPSR/studies/30263 (accessed on 30 October 2022).

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
