# Peer review of "The Dynamic Relationships between Educational Expectations and Science Learning Performance among Students in Secondary School and Their Later Completion of a STEM Degree"

_behavsci, 2024, doi:10.3390/bs14060506_

Round 1
Reviewer 1 Report
Comments and Suggestions for Authors
This article provides a significant contribution to our understanding of how to better educate future STEM professionals. The content is extremely relevant and it is a well-designed study. However, greater clarity is needed in the wording. Specific comments are included below in the order in which the issues appear in the paper.
Title, "relationships" needs to be capitalized.
Affiliation 2: the 2 is duplicated, once in superscript and once in normal script.
Abstract, first sentence "educational expectations of students" maybe add "for themselves" to clarify that it is not the expectations of teachers, parents, etc. instead.
Abstract, second sentence: inversely here is confusing. I normally interpret "inverse" to mean a negative correlation, not the other direction of a bi-directional relationship. Different wording would help.
Abstract, "current study investigated how..." sentence... the "which" should be "and."
Abstract: "national representative" should be "nationally representative." Should make clear in abstract this is a US context, as your affiliations are international and leave the location a mystery to the reader until the Method section.
Page 2, line 63: I think you need a comma here instead of the word "that."
Clause starting with "such as" on page 2, line 64: I think this would be resolved better with an em-dash, and some clear punctuation about where the clause ends. in general, your sentences are long and complex. Shortening them could clarify your arguments.
Page 2, line 68, "which:" it would help the grammar of this sentence if you begin this as a new sentence, e.g., "Thus, the current study argues that the science learning performance of students may possibly..."
Page 2, line 81, "that contribute:" this clause does not follow from the previous content of the sentence.
Page 2, line 89, "which are," I'm not sure what subject is doing the "are" verb here because of the convolutions of this sentence.
Theoretical framework of the Study: Capitalize "framework" to put the heading in sentence case capitalization. Further, this sentence is far too long with too many clauses. Separate it into multiple simpler sentences, e.g., "Cultivating STEM graduates and professionals is a progressive and cumulative process. It is strongly reliant on the educational development of students in secondary school. This period is formative in cultivating students' academic motivations, such as educational expectations. Secondary school also is a time for students to establish their essential science knowledge and skills for their future STEM development in adulthood." A broken up sentence like this would allow you to place the citations more specifically also, allowing the reader to know which piece of information from this complex, long sentence belongs to which authors. Additionally, you used "essential" and "essentially" twice just a few words apart at the end of the sentence. Take a similar approach to all of your sentences, identifying places where clauses can be separated into multiple independent sentences with clearer referents for pronouns and more apparent subject-predicate connections.
Lines 232-238: I wonder if you could simplify this section significantly by referring to "comparing students to each other" or "assessing between-subjects effects" and "comparing students' individual performance over time" or "within-subjects effects." You could make it much easier to parse for the reader.
Line 234: "her/his." We use "their" for the third-person singular in most style guides now.
Modelling Techniques section: I would like to see here a description of the difference between developmental and growth trajectories. You keep intentionally referring to both of those as if they are separate variables, but I don't know the difference. I wonder if other readers who do not regularly use growth curve modeling techniques may also struggle to understand.
Line 344, "contributed by..." do you mean "contributed to by their..."? Either way it's confusing wording. Try something instead like "the current study used parallel-process latent growth curve modelling (PP-LGCM) (citation) to use the developmental and growth trajectories of secondary students' educational expectations and science learning performance to predict their likelihood of graduation with a STEM degree in adulthood."
I like the inclusion and explanation of the equation.
Line 373: I don't think "ethnical" is a word.
Remove "as" from this sentence (like 373, after word "indicating"). Same for "referring as..." in same sentence. Should be "referring to."
Perhaps in modelling techniques or Results, you could explain what "cross-domain" correlations are. I do not have access to your tables, which may answer this concern: I am not sure.
"Native" in the context of Native American as an ethnicity should always be capitalized as a proper noun.
Beginning of discussion: Instead of "first of its attempts," try "first of its kind" or "first to attempt..." or even just "first to investigate" to clarify.
Line 574: "Are concerned by educators and policy makers" doesn't make sense, as "promotion of STEM development" (inanimate abstract concept) cannot be "concerned" (a human emotion). Instead, try something like "are of interest to educators and..."
Comments on the Quality of English Language
please see above comments
Author Response
For Reviewer 1
1) This article provides a significant contribution to our understanding of how to better educate future STEM professionals. The content is extremely relevant and it is a well-designed study. However, greater clarity is needed in the wording. Specific comments are included below in the order in which the issues appear in the paper.
Reply: Thank you, and now the whole study has been revised to make the study purpose, study relationships, methods and modeling procedures, findings, and discussion clearer and more succinct. In addition, the sentence structures and language presentations have been revised to be more accurate and concise.
2) Title, "relationships" needs to be capitalized.
Reply: The word "relationships" is now capitalized.
3) Affiliation 2: the 2 is duplicated, once in superscript and once in normal script.
Reply: It is corrected now.
4) Abstract, first sentence "educational expectations of students" maybe add "for themselves" to clarify that it is not the expectations of teachers, parents, etc. instead.
Abstract, second sentence: inversely here is confusing. I normally interpret "inverse" to mean a negative correlation, not the other direction of a bi-directional relationship. Different wording would help.
Reply: For the first sentence "educational expectations of students" in the Abstract, now it is re-written as "for themselves" to clarify that “Educational expectations of students for themselves have been commonly corroborated to directly predict their higher academic performance.” Besides, the term “inverse” refers to the inverse relationship between the developmental and growth trajectories of students’ educational expectations and science learning performance across the years of secondary school that is tested in the current study, which means to take the developmental and growth trajectories of students’ science learning performance as the predictors to predict the developmental and growth trajectories of students’ educational expectations. For the details of the direct and inverse PP-LGCM models, the current revised manuscript has been explained more clearly in the sections “2.1 The Direct Model from Educational Expectations of Students to Their Science Learning Performance and STEM Achievement” and “2.2 The Inverse Model from Science Learning Performance of Students to Their Educational Expectations and STEM Achievement” and “2.3. Modelling Techniques”.
5) Abstract, "current study investigated how..." sentence... the "which" should be "and."
Abstract: "national representative" should be "nationally representative." Should make clear in abstract this is a US context, as your affiliations are international and leave the location a mystery to the reader until the Method section.
Reply: They are corrected now.
6) Page 2, line 63: I think you need a comma here instead of the word "that."
Reply: The whole sentence has now been rewritten as “On the other hand, some limited short-term longitudinal research has reported that the relationship between educational motivation, such as academic engagement and self-efficacy, and students’ academic performance is reciprocally reinforced and influenced rather than unidirectionally from the former to the latter.”
7) Clause starting with "such as" on page 2, line 64: I think this would be resolved better with an em-dash, and some clear punctuation about where the clause ends. in general, your sentences are long and complex. Shortening them could clarify your arguments.
Reply: The whole sentence has now changed to “On the other hand, some limited short-term longitudinal research has reported that the relationship between educational motivation, e.g., academic engagement and self-efficacy, and students’ academic performance is reciprocally reinforced and influenced rather than unidirectionally from the former to the latter (Arens et al., 2017; Burns, Crisp, & Burns, 2020; Guo, Sun, Breit-Smith, Morrison, & Connor, 2015)," in which ‘e.g.,’ is used to replace 'such as’, and em-dash is not used in this sentence due to ‘educational motivation’ covers not only “academic engagement and self-efficacy” but also other related constructs. In this sentence, “academic engagement and self-efficacy” is just an example to indicate that those common constructs of educational motivation are used in research to predict the academic performance and achievement of students. In the current study, the educational expectations of students are used to predict their STEM achievement. In addition, the whole manuscript has now been revised to make its sentences more precise, clearer, and accurate.
8) Page 2, line 68, "which:" it would help the grammar of this sentence if you begin this as a new sentence, e.g., "Thus, the current study argues that the science learning performance of students may possibly..."
Reply: The structure of the sentences is now revised to “On the other hand, some limited short-term longitudinal research has reported that the relationship between educational motivation, e.g., academic engagement and self-efficacy, and students’ academic performance is reciprocally reinforced and influenced rather than unidirectionally from the former to the latter (Arens et al., 2017; Burns, Crisp, & Burns, 2020; Guo, Sun, Breit-Smith, Morrison, & Connor, 2015). This denotes in the current study that the science learning performance of students may possibly shape students’ development and changes in educational expectations in secondary school, which then jointly contribute to students’ future educational achievement, such as successful graduation with a STEM degree in adulthood.” I hope this revision can help present the meanings of the inverse relationship between students’ educational expectations and science learning performance in a clearer and more precise way.
9) Page 2, line 81, "that contribute:" this clause does not follow from the previous content of the sentence.
Reply: The whole sentence structure is now changed to “Due to the paucity of research investigating how the dynamic and changeable processes of students’ educational motivation and academic performance may shape each other over time and then later contribute to the educational achievement of students, the current study is intended to examine the reciprocal relationship between the developmental and growth trajectories of students’ educational expectations and science learning performance in secondary school and analyze how they may jointly predict students’ later successful completion of a STEM degree in adulthood.” I hope this new presentation will be clearer and more accurate.
10) Page 2, line 89, "which are," I'm not sure what subject is doing the "are" verb here because of the convolutions of this sentence.
Reply: The whole sentence is now revised and broken down to “Findings of this study can help unfold the long-term and complex reciprocity of students’ early educational motivation and academic performance in the transitional period of secondary school years in connection with their later educational success in adulthood. This is conducive to educational reforms and policy designs, especially for science education and students’ STEM development.” I hope the meanings can be more accurate and structural.
11) Theoretical framework of the Study: Capitalize "framework" to put the heading in sentence case capitalization. Further, this sentence is far too long with too many clauses. Separate it into multiple simpler sentences, e.g., "Cultivating STEM graduates and professionals is a progressive and cumulative process. It is strongly reliant on the educational development of students in secondary school. This period is formative in cultivating students' academic motivations, such as educational expectations. Secondary school also is a time for students to establish their essential science knowledge and skills for their future STEM development in adulthood." A broken up sentence like this would allow you to place the citations more specifically also, allowing the reader to know which piece of information from this complex, long sentence belongs to which authors. Additionally, you used "essential" and "essentially" twice just a few words apart at the end of the sentence. Take a similar approach to all of your sentences, identifying places where clauses can be separated into multiple independent sentences with clearer referents for pronouns and more apparent subject-predicate connections.
Reply: Now the whole manuscript has been revised and proofread to make the structures of the sentences more concise and succinct in meaning. For example, the long sentence of “Cultivating STEM graduates and professionals is a progressive and cumulative process that is strongly reliant on the educational development of students in secondary school, which is the formative period to cultivate students’ academic motivations, such as educational expectations, and establish their essential science knowledge and skills essentially for their future STEM development in adulthood (Herskovic & Silva, 2022; Miller & Pearson, 2012; Penprase, 2020)” has now been re-written to “Cultivating STEM graduates and professionals is a progressive and cumulative process that is critically reliant on the formative years of secondary school for students to establish their educational motivation, e.g., educational expectations, and cultivate their essential science knowledge and skills adequately (Herskovic & Silva, 2022; Miller & Pearson, 2012; Penprase, 2020).” In addition, the word “essentially” has been deleted.
12) Lines 232-238: I wonder if you could simplify this section significantly by referring to "comparing students to each other" or "assessing between-subjects effects" and "comparing students' individual performance over time" or "within-subjects effects." You could make it much easier to parse for the reader.
Reply: The principles and meanings of the developmental and growth trajectories of students’ educational expectations and science learning performance across the years of secondary school being modeled by PP-LGCM modelling are not exactly equivalent to the definitions “between-subjects effects", "within-subjects effects”, and "comparing students' individual performance over time". Hence, they are not replaceable. Nevertheless, I have tried to present the meanings clearly, and I expect that readers of a refereed academic journal should be academically equipped adequately to read an academic or research papers in the field(s) they are interested in. In fact, I have attempted my best to present the use of PP-LGCM modeling in its simplest way compared to existing published peer-reviewed papers:
Kristensen, S. M., Danielsen, A. G., Urke, H. B., Larsen, T. B., & Aanes, M. M. (2023). The positive feedback loop between academic self-efficacy, academic initiative, and Grade Point Average: a parallel process latent growth curve model. Educational Psychology, 43(7), 835-853. doi:10.1080/01443410.2023.2242603
Lau, M. A., Temcheff, C. E., Poirier, M., Commisso, M., & Déry, M. (2023). Longitudinal relationships between conduct problems, depressive symptoms, and school dropout. Journal of School Psychology, 96, 12-23. doi:10.1016/j.jsp.2022.10.005
Tvedt, M. S., Virtanen, T. E., & Bru, E. (2022). Trajectory subgroups of perceived emotional support from teachers: Associations with change in mastery climate and intentions to quit upper secondary school. Learning and Instruction, 80, 101562. doi:10.1016/j.learninstruc.2021.101562
13) Line 234: "her/his." We use "their" for the third-person singular in most style guides now.
Reply: Now "her/his" is changed to "their”.
14) Modelling Techniques section: I would like to see here a description of the difference between developmental and growth trajectories. You keep intentionally referring to both of those as if they are separate variables, but I don't know the difference. I wonder if other readers who do not regularly use growth curve modeling techniques may also struggle to understand.
Reply: The description and the definitions of developmental and growth trajectories of students’ educational expectations and science learning performance, plus completion of a STEM degree in adulthood, are elaborated in ‘2.3 The Present Study’, which are written:
“In testing the above-proposed longitudinal causal relationship dynamically, the developmental and growth trajectories of students’ educational expectations and science learning performance are considered as the developmental trend of a student relative to their classmates in a standing track in the years of secondary school (the development of educational expectations and science learning performance) and the intrapersonal changes of this student during the years of secondary school (the growth of educational expectations and science learning performance) captured latently by repeated measurements from grade 8 to grade 12. In addition, students’ successful completion of a STEM degree in adulthood indicates whether the student participant graduated with a four-year baccalaureate degree majoring in science, technology, engineering, mathematics, or medicine (STEMM) or not, which is a common definition adopted in STEM research.” (Refer to the second paragraph in the section of ‘2.3 The Present Study’).
In fact, the developmental and growth trajectories of students’ educational expectations and science learning performance are two different latent factors, in which the former represents the initial development level of the students in secondary school and the latter means the changes of the students in educational expectations and science learning performance across time.
15) Line 344, "contributed by..." do you mean "contributed to by their..."? Either way it's confusing wording. Try something instead like "the current study used parallel-process latent growth curve modelling (PP-LGCM) (citation) to use the developmental and growth trajectories of secondary students' educational expectations and science learning performance to predict their likelihood of graduation with a STEM degree in adulthood."
I like the inclusion and explanation of the equation.
Reply: Now, "contributed by..." is changed to "contributed to by”. In addition, to more clearly and succinctly explaining how to use the PP-LGCM modeling to examine the longitudinal causal relationship in the current study, the original sentences in the section “3.3 Modeling Techniques” are rewritten to:
“To predict the outcome of students’ successful completion of a STEM degree in adulthood contributed to by their developmental and growth trajectories of educational expectations and science learning performance in secondary school, parallel-process latent growth curve modeling (PP-LGCM) was conducted (Smid, Depaoli, & Van De Schoot, 2020). PP-LGCM is an extended type of growth modeling that is reliable and flexible for analyzing longitudinal casual relationships by their latent trajectories in connection to a distal outcome. In the growth modeling framework, latent variables are estimated for the initial developmental levels (intercepts) and the changes (slopes) occurring over time that are used to represent the developmental and growth trajectories of students’ educational expectations and science learning performance during the years of secondary school from grade 8 to grade 12 in the current study.” (Refer to the first paragraph of “3.3 Modeling Techniques”).
In addition, the equations of parallel-process latent growth curve modeling (PP-LGCM) are explained more in detail now in the section “3.3 Modeling Techniques” for readers to have more information about the modeling procedures of using PP-LGCM techniques to test the longitudinal causal relationship of the current study.
16) Line 373: I don't think "ethnical" is a word.
Reply: Now, the word "ethnical" is corrected.
17) Remove "as" from this sentence (like 373, after word "indicating"). Same for "referring as..." in same sentence. Should be "referring to."
Reply: "as" has been removed, and “"referring as..." has been changed to "referring to."
18) Perhaps in modelling techniques or Results, you could explain what "cross-domain" correlations are. I do not have access to your tables, which may answer this concern: I am not sure.
Reply: Cross-domain correlations mean the correlations of the different study variables measured repeatedly over time, and the same-domain correlations refer to the correlations of the same study variables measured repeatedly over time. I think, as a peer-reviewed study published in a refereed academic journal, readers should have adequate scholarly training to understand the statistical modeling techniques used in the current study.
19) "Native" in the context of Native American as an ethnicity should always be capitalized as a proper noun.
Reply: Now “Native American” or “Native Americans” have been used throughout the whole manuscript.
20) Beginning of discussion: Instead of "first of its attempts," try "first of its kind" or "first to attempt..." or even just "first to investigate" to clarify.
Reply: Now, "first of its attempts" has been changed to "first of its kind".
21) Line 574: "Are concerned by educators and policy makers" doesn't make sense, as "promotion of STEM development" (inanimate abstract concept) cannot be "concerned" (a human emotion). Instead, try something like "are of interest to educators and..."
Reply: The presentations in Section “5. Discussion” have been substantially revised, in which “Are concerned by educators and policy makers" has now been changed to:
“Furthermore, as science education and promotion of students’ STEM development are of interest to educators and policy makers because of their profound implications for the cultivation of STEM intellectuals to promote societal, economic, cultural, and technological advancements (Anderson & Li, 2020; Miller & Pearson, 2012; Wright, Ellis, & Townley, 2017), more research is needed to scrutinize how the developmental and growth trajectories of students’ educational expectations and science learning performance during the course of secondary school may affect students’ later STEM success in adulthood.”
I hope this completely revised version of the manuscript can make the presentation of the study purpose, study relationships, research methods, results, and discussion of the study more accurate, precise, and understandable.
Reviewer 2 Report
Comments and Suggestions for Authors
The study presents a strong literature review.
The study's hypotheses are clearly formulated and well linked to answer the study's purpose.
About Methodology (Sample and Data): Wouldn't it be important to know more about the backgrounds of the study participants when they attended secondary school?
It would be good if the conclusion were to take up the hypotheses formulated. Confirmed or refuted?
They are confirmed, but it would be good to state them explicitly
Author Response
# Reviewer 2
Open Review
(x) I would not like to sign my review report
( ) I would like to sign my review report
Quality of English Language
(x) I am not qualified to assess the quality of English in this paper
( ) English very difficult to understand/incomprehensible
( ) Extensive editing of English language required
( ) Moderate editing of English language required
( ) Minor editing of English language required
( ) English language fine. No issues detected
|
Yes |
Can be improved |
Must be improved |
Not applicable |
|
|
Is the content succinctly described and contextualized with respect to previous and present theoretical background and empirical research (if applicable) on the topic? |
(x) |
( ) |
( ) |
( ) |
|
Are all the cited references relevant to the research? |
(x) |
( ) |
( ) |
( ) |
|
Are the research design, questions, hypotheses and methods clearly stated? |
(x) |
( ) |
( ) |
( ) |
|
Are the arguments and discussion of findings coherent, balanced and compelling? |
( ) |
(x) |
( ) |
( ) |
|
For empirical research, are the results clearly presented? |
(x) |
( ) |
( ) |
( ) |
|
Is the article adequately referenced? |
(x) |
( ) |
( ) |
( ) |
|
Are the conclusions thoroughly supported by the results presented in the article or referenced in secondary literature? |
( ) |
(x) |
( ) |
( ) |
Comments and Suggestions for Authors
R2.1) The study presents a strong literature review.
Reply: Thank you.
R2.2) The study's hypotheses are clearly formulated and well linked to answer the study's purpose.
Reply: Thank you.
R2.3) About Methodology (Sample and Data): Wouldn't it be important to know more about the backgrounds of the study participants when they attended secondary school?
Reply: Yes, as the longitudinal data was obtained from the United States, where there is an ethnically and family-diverse society; hence, the background information is important for readers to know the sociodemographic characteristics of the student participants that may affect their educational expectations and science learning performance, as well as their later completion of a STEM degree.
R2.4) It would be good if the conclusion were to take up the hypotheses formulated. Confirmed or refuted?
Reply: Agree. The section of Conclusion has now been revised to add the following contents to summarize the findings and hypotheses:
“The current study supported the argument that students’ educational expectations and their science learning performance in the years of secondary school education are developmental and changing in nature, in which the growth trajectories of students’ educational expectations and science learning performance are a function of their developmental trajectories in the same domain, and they are also mutually reinforced across time to contribute to students’ later successful completion of a STEM degree in adulthood. The above-mentioned longitudinal causal relationships between the developmental and growth trajectories of students’ educational expectations and science learning performance and students’ later STEM achievement are vindicated even after adjusting for the influential sociodemographic covariates of students’ gender, family composition, and ethnicity. In fact, the external validity of the findings in the current study is fortified by accounting for the nestedness of the clustered data structure of LSAY at the school level. Nevertheless, there are some limitations existing in the current study that are needed to be improved in future research.”
Reviewer 3 Report
Comments and Suggestions for Authors
I recommend author the following suggestions to improve the clarity of the content:
1- The introduction can be separated into 2 sections: There can be a brief introduction addressing the rationale of investigating the association among educational expectations and science performance. And the second section will be the related research.
2-I suggest author to discuss hypothesis related studies first and then after neath present each hypothesis to improve flow.
3- some terms require clear definition. For instance; "developmental and growth trajectories" please give thorough definition of the term. It seems like measuring 2 different concepts.
4- I 'm not sure how to author measured "developmental and growth trajectories" of students
5- Another issue is there is NO TABLE inside the content although the content refers to specific table captions in the appendix section. There is NO APPENDIX in the manuscript. Therefore I can not verify the appropriateness of the table information. Please add them in the revised version.
6- The dataset subject to this study is from 2014 form USA. Author should address why such outdated dataset were used. It has ben almost a decade past. Eventually the educational expectations of earlier generation and today's generation are not the same. Please address the limitations of the study.
7- There is some information about the statistical analysis used in the study i.e. parallel-process latent growth curve modelling (PP-LGCM). could author give specific studies applying this statistical technique to similar attributes. Is there an alternative technique to answer hypotheses?
8- Please divide your discussion with respect to each hypothesis of the study.
9- The conclusion includes limitations rather than the conclusion. could the researcher strengthen the outcomes of the study.
10- what are the opinions about the practical implications of such study. It might be wiser to address implications.
11-In Figure 1, there is a correlation between educational expectations at grade 9 and grade 10 and science learning at grade 9 and grade 10. Also in Figure 2, there are correlations between educational expectations at grade 9 and grade 10 and between grade 10 and grade 11. How did the author decide the correlations among the variables?
13- There are 1-2 recent references. Could the author aim for locating recent relevant sources of the study.
Comments on the Quality of English Language
Proofreading may be required.
Author Response
Reviewer 3
Open Review
(x) I would not like to sign my review report
( ) I would like to sign my review report
Quality of English Language
( ) I am not qualified to assess the quality of English in this paper
( ) English very difficult to understand/incomprehensible
( ) Extensive editing of English language required
(x) Moderate editing of English language required
( ) Minor editing of English language required
( ) English language fine. No issues detected
|
Yes |
Can be improved |
Must be improved |
Not applicable |
|
|
Is the content succinctly described and contextualized with respect to previous and present theoretical background and empirical research (if applicable) on the topic? |
( ) |
( ) |
(x) |
( ) |
|
Are all the cited references relevant to the research? |
( ) |
(x) |
( ) |
( ) |
|
Are the research design, questions, hypotheses and methods clearly stated? |
( ) |
( ) |
(x) |
( ) |
|
Are the arguments and discussion of findings coherent, balanced and compelling? |
( ) |
( ) |
(x) |
( ) |
|
For empirical research, are the results clearly presented? |
( ) |
( ) |
(x) |
( ) |
|
Is the article adequately referenced? |
( ) |
( ) |
(x) |
( ) |
|
Are the conclusions thoroughly supported by the results presented in the article or referenced in secondary literature? |
( ) |
( ) |
(x) |
( ) |
Comments and Suggestions for Authors
I recommend author the following suggestions to improve the clarity of the content:
R3.1) 1- The introduction can be separated into 2 sections: There can be a brief introduction addressing the rationale of investigating the association among educational expectations and science performance. And the second section will be the related research.
Reply: The section of ‘1. Introduction’ has now been revised, which includes two parts to first address the rationale of the study and then introduce the importance of the current study and its purpose by referring to related research.
R3.2) 2-I suggest author to discuss hypothesis related studies first and then after neath present each hypothesis to improve flow.
Reply: This study has in fact discussed the hypothesis-related studies first and then presented the hypotheses afterward. The difference is that I put the three main hypotheses of the study in the section of ‘2.3 The Present Study’ to integrate them together for better understanding the complex study relationships presented in the current study for readers.
R3.3) 3- some terms require clear definition. For instance; "developmental and growth trajectories" please give thorough definition of the term. It seems like measuring 2 different concepts.
Reply: The definitions and meanings of the developmental and growth trajectories of students’ educational expectations and science learning performance are clearly defined in the section “2.3 The Present Study’, which has written:
“More important and pertinent, some scholars have recently proposed to take a dynamic approach to study the educational motivation, including educational expectations, and academic performance of students across times to see how their developmental and changing processes affect students’ later educational achievement (Arens et al., 2017; Marsh, 2022; O'Donnell, Redmond, Thomson, Wang, & Turkmani, 2022). In fact, more advanced but limited recent longitudinal studies have reported that educational motivation and academic performance of students are an adaptation process that is developmental and changeable across times rather than fixed and stable (Marsh, 2022; Zhao, Li, Ma, & Zhang, 2019).” (First paragraph in the section ‘2.3 The Present Study’)
“In testing the above-proposed longitudinal casual relationship dynamically, the developmental and growth trajectories of students’ educational expectations and science learning performance are considered as the developmental trend of a student relative to their classmates in a standing track in the years of secondary school (the development of educational expectations and science learning performance) and the intrapersonal changes of this student during the years of secondary school (the growth of educational expectations and science learning performance) captured latently by repeated measurements from grade 8 to grade 12.” (Second paragraph in the section ‘2.3 The Present Study’)
R3.4) 4- I 'm not sure how to author measured "developmental and growth trajectories" of students
Reply: As responded in R3.3, the developmental and growth trajectories of students’ educational expectations and science learning performance are defined in the current study as:
“In testing the above-proposed longitudinal casual relationship dynamically, the developmental and growth trajectories of students’ educational expectations and science learning performance are considered as the developmental trend of a student relative to their classmates in a standing track in the years of secondary school (the development of educational expectations and science learning performance) and the intrapersonal changes of this student during the years of secondary school (the growth of educational expectations and science learning performance) captured latently by repeated measurements from grade 8 to grade 12.” (Second paragraph in the section of ‘2.3 The Present Study’)
3.5) 5- Another issue is there is NO TABLE inside the content although the content refers to specific table captions in the appendix section. There is NO APPENDIX in the manuscript. Therefore I can not verify the appropriateness of the table information. Please add them in the revised version.
Reply: The Tables and the appendix, as well as the figures, are inserted in the reviewing version of the manuscript by Behavioral Sciences.
R3.6) 6- The dataset subject to this study is from 2014 form USA. Author should address why such outdated dataset were used. It has ben almost a decade past. Eventually the educational expectations of earlier generation and today's generation are not the same. Please address the limitations of the study.
Reply: Agree, the study has addressed this issue of the dataset that may be too long from now; hence, the limitations have addressed this concern:
“Nevertheless, there are some limitations existing in the current study that are needed to be improved in future research. First, the data from LSAY were collected thirty years ago, starting in 1987 and ending in 2011, for which the learning attitudes and science education have apparently changed (Anderson & Li, 2020). Hence, newer and updated data are needed to further confirm the study relationships found in the current study.” (in the section of ‘Conclusion’)
R3.7) 7- There is some information about the statistical analysis used in the study i.e. parallel-process latent growth curve modelling (PP-LGCM). could author give specific studies applying this statistical technique to similar attributes. Is there an alternative technique to answer hypotheses?
Reply: The PP-LGCM modeling has been recently used in educational, developmental, and behavioral science research for investigating more advanced longitudinal causal relationships that are hard in the past to be examined, albeit having been conceptualized by academics. Hence, it is currently absence of any alternatives to answer the hypotheses set in the current study as the developmental and growth trajectories of students’ educational expectations and science learning performance are reciprocally reinforced and mutually affected in a latent sense. For this, I have cited some recent related research studies in the section “3.3. Modeling Techniques” as references for readers to refer to:
“This type of modeling procedure for the study of longitudinal causal relationships has been recently used in educational, developmental, and behavioral science research (Kristensen, Danielsen, Urke, Larsen, & Aanes, 2023; Tvedt, Virtanen, & Bru, 2022).” (Refer to second paragraph of “3.3. Modeling Techniques”)
Kristensen, S. M., Danielsen, A. G., Urke, H. B., Larsen, T. B., & Aanes, M. M. (2023). The positive feedback loop between academic self-efficacy, academic initiative, and Grade Point Average: A parallel process latent growth curve model. Educational Psychology, 43(7), 835-853. doi:10.1080/01443410.2023.2242603
Tvedt, M. S., Virtanen, T. E., & Bru, E. (2022). Trajectory subgroups of perceived emotional support from teachers: Associations with change in mastery climate and intentions to quit upper secondary school. Learning and Instruction, 80, 101562. doi:10.1016/j.learninstruc.2021.101562
R3.8) 8- Please divide your discussion with respect to each hypothesis of the study.
Reply: Now the section “5. Discussion” has elaborated according to the findings of the current study based on the hypotheses set.
R3.9) 9- The conclusion includes limitations rather than the conclusion. could the researcher strengthen the outcomes of the study.
Reply: Now the ‘6. Conclusion’ has been strengthened by summarizing the main findings of the current study:
“The current study supported the argument that students’ educational expectations and their science learning performance in the years of secondary school education are developmental and changing in nature, in which the growth trajectories of students’ educational expectations and science learning performance are a function of their developmental trajectories in the same domain, and they are also mutually reinforced across time to contribute to students’ later successful completion of a STEM degree in adulthood. The above-mentioned longitudinal causal relationships between the developmental and growth trajectories of students’ educational expectations and science learning performance and students’ later STEM achievement are vindicated even after adjusting for the influential sociodemographic covariates of students’ gender, family composition, and ethnicity. In fact, the external validity of the findings in the current study is fortified by accounting for the nestedness of the clustered data structure of LSAY at the school level. Nevertheless, there are some limitations existing in the current study that are needed to be improved in future research.”
R3.10) 10- what are the opinions about the practical implications of such study. It might be wiser to address implications.
Reply: The implications have been introduced in the section of ‘5. Discussion’ by relating to the findings of the current study. (Refer to the section "5. Discussion")
R3.11) 11-In Figure 1, there is a correlation between educational expectations at grade 9 and grade 10 and science learning at grade 9 and grade 10. Also in Figure 2, there are correlations between educational expectations at grade 9 and grade 10 and between grade 10 and grade 11. How did the author decide the correlations among the variables?
Reply: The decision to make covariances between the residuals is explained in Section ‘4. Results’:
“Table 3 shows the results of the univariate direct PP-LGCM model investigating the effects of students’ educational expectations on students’ science learning performance in secondary school (PP-LGCM Model 1A), for which a very good model fit was obtained: CFI =.972; RMSEA =.062; SRMR =.048. Nevertheless, the modification indices suggested setting the covariances between the residuals of students’ grade-9 and grade-10 educational expectations, grade-9 and grade-10 science learning performance, and grade-10 and grade-11 science learning performance, in which an excellent model fit appeared: CFI =.984; RMSEA =.048; SRMR =.039.” (Paragraph 3 in the section of ‘3. Results’)
“On the other side, the univariate inverse PP-LGCM model was conducted to test the effects of students’ developmental and growth trajectories of science learning performance on their developmental and growth trajectories of educational expectations in secondary school (PP-LGCM Model 2A). Table 4 shows that an excellent model fit was obtained: CFI= .974; RMSEA= .059; SRMR= .044. Nevertheless, the modification indices recommended setting covariances between the residuals of students’ grade-9 and grade-10 as well as grade-10 and grade-11 science learning performance, for which a better excellent model fit was obtained: CFI= .987; RMSEA= .043; SRMR= .033.” (Paragraph 5 in the section of ‘3. Results’)
R3.12) 13- There are 1-2 recent references. Could the author aim for locating recent relevant sources of the study.
Reply: Agree; now the revised version has cited more recent relevant references.
Round 2
Reviewer 3 Report
Comments and Suggestions for Authors
Thank you for the clarification of the suggested issues. I believe that the author made a remarkable effort to clarify the issues addressed in the earlier review.
The only addition would be adding practical implications of the study as a separate section.
Comments on the Quality of English Language
The English of the content was improved in the revised version.
Author Response
Dear Ms. Marcia Min:
Thank you for sending me the second-round review comments, I have revised the manuscript according to the comments of reviewer 3, and my responses are below:
R3.1) The only addition would be adding practical implications of the study as a separate section.
Reply: Practical implications have been added in the section of Conclusion as I think the section of ‘6. Conclusion’ is to summarize the study findings of the current study, which is fitted to carry out the practical implications of the study that are written as:
“The current study supported the argument that students’ educational expectations and their science learning performance in the years of secondary school education are developmental and changing in nature, in which the growth trajectories of students’ educational expectations and science learning performance are a function of their developmental trajectories in the same domain, and they are also mutually reinforced across time to contribute to students’ later successful completion of a STEM degree in adulthood. The above-mentioned longitudinal causal relationships between the developmental and growth trajectories of students’ educational expectations and science learning performance and students’ later STEM achievement are vindicated even after adjusting for the influential sociodemographic covariates of students’ gender, family composition, and ethnicity. In fact, the external validity of the findings in the current study is fortified by accounting for the nestedness of the clustered data structure of LSAY at the school level. Apparently, this study supported that both the educational expectations and science learning performance of students during the years of secondary school are developmental and changeable, which means that these important educational elements of students in the early years are cultivable and malleable. Educators and policymakers should hence support adolescent students to establish and maintain their higher educational expectations and better science learning performance in an ongoing process. This is important as they both affect whether the students can succeed in STEM development in adulthood. In addition, the reciprocal and bidirectional relationship between students’ educational expectations and science learning performance was corroborated in the current study. Hence, the academic motivation of students and their educational performance during secondary school years should be monitored and supported by teachers, educators, and youth practitioners to help them have better development and sustenance in parallel because they are influential on students’ trajectories to become STEM professionals in the future. Furthermore, as the sociodemographic covariates of students, e.g., gender, family composition, and ethnicity, affect their developmental and growth trajectories of educational expectations and science learning performance as well as STEM achievement in adulthood, responsive and effective educational and intervention strategies and initiatives should be introduced to support the STEM development of students with different sociodemographic characteristics.” (First paragraph in the section of ‘6. Conclusion and Implications’)
R3.2) The English of the content was improved in the revised version.
Reply: Yes, the English language has been proofread in the re-submitted version of first round review, and now I have read again the manuscript to ensure the fluency and accuracy of its language structure.
Best
The author